# Comparative Study on the Behavior of Keyhole in Analogy Welding and Real Deep Penetration Laser Welding

**DOI:** 10.3390/ma15249001

**Published:** 2022-12-16

**Authors:** Zhongjia Hao, Huiyang Chen, Xiangzhong Jin, Zuguo Liu

**Affiliations:** State Key Laboratory of Advanced Design and Manufacturing for Vehicle Body, Hunan University, Changsha 410082, China

**Keywords:** analogy, deep penetration laser welding, keyhole, water model, sandwich method

## Abstract

In deep penetration laser welding, the behavior of the keyhole has an important influence on the welding quality. As it is difficult to directly observe the keyhole and detect the pressure inside the keyhole during metal laser welding, theoretical analysis and numerical simulation methods are commonly used methods in studying keyhole behavior. However, these methods cannot provide direct real information on keyhole behavior. In this paper, a method of analogy welding is proposed, in which high speed gas is used to blow the liquid to generate the keyhole. Relevant process experiments were conducted to explore keyhole behavior in analogy welding and real deep penetration laser welding. The pressure balance of the keyhole, both in analogy welding and real deep penetration laser welding, were analyzed. The laws obtained in analogy welding and real deep penetration laser welding are similar, which indicates that studying keyhole formation and the maintenance principle using the analogy welding method proposed in this paper may be helpful for deep understanding of the keyhole formation and maintenance mechanisms in real deep penetration laser welding.

## 1. Introduction

As an advanced materials processing technology, laser welding has been used in a wide range of scientific and industrial applications, e.g., steel, nuclear power, aerospace, rail transit, automotive, electronics and other major civil or military projects. It plays an increasingly important role in lightweight transportation vehicles by the welding of thin-wall components, Al–Mg alloy and other light materials. Metal materials, such as aluminum alloy and steel, are widely used in the industrial field, and they are the main objects of laser welding. Generally, the modes of laser welding are mainly conduction mode and keyhole mode. When the laser energy is insufficient to evaporate the material, that is, the maximum temperature does not exceed the boiling point of the material, this welding mode is called the conduction mode. When the laser energy is enough to evaporate the material, a keyhole appears inside the material, and, thus, this welding mode is called the keyhole mode. Keyhole mode welding, or deep penetration laser welding, has the advantages of high aspect ratio, high utilization of laser energy and high welding efficiency, and it is the main research direction in the welding field at present. For commonly used aluminum alloy, when the laser power is 3.2 kW and the welding speed is 20 mm/s, deep penetration laser welding can be carried out on 5052 aluminum alloy plates with a thickness of 4mm, and the weld surface is bright and smooth. Of course, deep penetration laser welding also has some limitations, such as high requirements for workpiece assembly, high processing costs, and limited welding depth, especially compared with the depth of electron beam welding.

In deep penetration laser welding, the formation of a keyhole indicates that the coupling mode of laser energy and material has changed; that is, the laser energy absorption has changed from material surface absorption to material interior absorption. The formation and maintenance of the keyhole is the key to realizing deep penetration welding. At present, there are still many defects in deep penetration laser welding, such as weld depression, spatter and multi-humps, which hinder the further development of this technology. The occurrence of defects is closely related to the keyhole behavior caused by the melting, evaporation and flow of the molten pool during the welding process. Therefore, it is of great significance to study the formation and maintenance mechanisms of the keyhole in deep penetration laser welding so as to optimize the welding process and improve the weld quality. However, it is difficult to directly observe the complete keyhole shape inside the metal, due to the opacity of metal, and the pressure balance inside the keyhole is even more difficult to detect directly.

### 1.1. Keyhole Observation and Theoretical Research

Many scholars have made achievements in keyhole observation and theoretical research on energy and mechanics in keyholes. Arata et al. [1] tried to capture the side shape of the keyhole from the vertical direction of the welding speed using X-ray, and directly observed the fluctuation of the keyhole. Dowden et al. [2,3,4,5,6] successfully built theoretical models of the open keyhole and the blind keyhole, studied the energy balance and pressure balance in the keyhole during the formation and maintenance of the keyhole, explored the generation of thermal stress during laser welding of thin plates, determined mathematically that the viscous resistance related to steam movement was one of the factors causing the violent movement of the keyhole and molten pool, and established a mathematical model to describe the axial movement of the molten pool along the keyhole wall. Semak et al. [7,8] took pictures of the keyhole and the molten pool with a high- speed camera. It was calculated that the metal vapor induced by the laser was significantly larger than the surface tension and static pressure of the molten pool, thus promoting the flow deformation of the molten pool to generate the keyhole. When the keyhole reached a certain size, it would gradually close under the action of surface tension and static pressure, and the process showed a certain periodicity. Kim et al. [9] used a high-speed camera to observe the influence of zinc coating on the morphology of the keyhole in the laser welding process. It was found that, due to the existence of the zinc coating, under the strong evaporation of zinc, the bottom of the keyhole was basically open, while the bottom of the keyhole of the ungalvanized steel tended to close. This phenomenon shows that when considering the pressure balance in the keyhole, the influence caused by the evaporation of elements contained in the material needs to be carefully considered. In order to estimate the keyhole depth in the actual welding process, Lankalapalli et al. [10] established a penetration model of a two-dimensional conical keyhole and linked the penetration depth with the incident power. Solana et al. [11] established an axisymmetric model of multiple reflections inside the keyhole, and ensured the model took into account the inverse bremsstrahlung absorption. In Pecharapa’s study [12], the phase transformation of materials in the welding process was taken into account, and a relatively good theoretical model was obtained. In the study by Strömbeck [13], the multiple reflection of the laser in the keyhole was described as the self-focusing of the laser welding system, and a model with higher temperature at the bottom of the keyhole was obtained. Fabbro and Chouf [14] investigated the uniform motion of the keyhole along a straight line, considered the multiple reflection and inverse bremsstrahlung absorption inside the keyhole, and linked the drilling rate of the laser beam with the moving speed of the keyhole. Then, a high-speed camera was used to observe the flow of the molten pool, and it was found that the interaction between the steam plume and the molten pool was the reason for the change of flow characteristics [15,16,17]. In the experiment, the welding speed was changed from slow to fast, and it was found that the inverse of the keyhole depth and the welding speed were almost linear [18].

With in-depth study of the keyhole, researchers can obtain a clear image of the keyhole in the laser welding process. The results obtained from direct observation experiments of the keyhole have become the basis for studying the characteristics of the keyhole. Jin’s team used the sandwich method to directly observe the keyhole [19,20,21,22], and constructed a three-dimensional multi reflection model of the keyhole. It was found that most of the positions of the rear wall of the keyhole were not irradiated by the laser. In order to maintain the energy balance inside the keyhole, the energy required for the rear wall of the keyhole would be transmitted from the front wall to the rear wall by the molten pool flow [23,24,25]. By using the sandwich method, Cheng et al. [26,27] found that, compared with Fresnel absorption, the inverse bremsstrahlung absorption of laser energy by keyhole plasma played a major role in absorbing laser energy, and the electron temperature inside the keyhole was uneven and distributed in the radial and depth directions. Li et al. [28] observed that the steam flowing upward and downward inside the keyhole formed a steam vortex after meeting, and the fluctuation of steam flow and pressure were the key factors leading to the fluctuation of the keyhole. Zhang et al. [29] used the modified sandwich method to observe the keyhole, and summarized the formation of the keyhole into three stages: the fast-drilling stage, the slow-drilling stage, and the quasi-steady state stage. They believe that the key factor to make the width of the molten pool enter the quasi-steady state was the balance between the rotation of the vortex and the lateral flow around the keyhole. Others believe that the steam recoil pressure generated by the energy reflected from the front wall of the keyhole to the rear wall is the main driving force for the deformation of the rear wall of the keyhole, and the rear wall collapses due to large surface tension and hydrostatic pressure during the oscillation of the keyhole [30].

### 1.2. Numerical Simulation of Keyhole

Numerical simulation is used to study the characteristics of keyholes. Wang et al. [31] established a three-dimensional heat source model, composed of a rotating Gaussian volume heat source and a double ellipsoidal heat source, to simulate the keyhole in the laser welding process. The numerical simulation results showed that the eddy currents formed at the top and bottom of the weld pool were conducive to the overall heat transfer. Cho et al. [32] simulated the flow of the molten pool at the initial stage of the keyhole formation, and found that, in the initial stage, the flow direction in the center of the molten pool appeared as axisymmetric oscillation, which was closely related to the recoil pressure, the cooling and the surface tension. Huang et al. [33] theoretically studied the correlations between the surface area, the volume of the keyhole, the welding speed and the surface tension coefficient, and believed that the surface tension controlled the oscillation period of the keyhole. Bedenko et al. [34] conducted one-dimensional simulation research on the dynamics of keyhole plasma during laser welding, and pointed out that the keyhole plasma had a periodic shielding effect on laser radiation, which caused the absorption of laser energy by workpiece materials to alternately attenuate or stop, resulting in the pressure and temperature oscillations. Pang et al. [35] proposed a mathematical model to describe the dynamic coupling behavior of the keyhole and the weld pool, and pointed out that the surface tension had a great influence on the period of keyhole depth oscillation. The oscillation of the plume ejected from the keyhole was closely related to the instability of the keyhole, and the oscillation frequency was the same as the oscillation period of the keyhole in the depth direction. Li et al. [36] simulated the laser welding of aluminum alloy under sub-atmospheric pressure. The numerical results showed that, compared with atmospheric pressure, the keyhole became wider and deeper, and the hump smaller. With the decrease of environmental pressure, the eddy current on the rear wall of the keyhole decreased or even disappeared, which was conducive to improving the stability of deep penetration laser welding and suppressing the generation of defects. Cunningham, Mayi studied the transformation of the welding mode, and found that there was a clear threshold for the sudden change from the conduction mode to the keyhole mode. During the transformation process, a semicircular pit appeared in the weld pool. Its depth and energy balance were determined by the effect of the recoil pressure on the weld pool [37,38]. Zou et al. [39] believed that during the welding process only the front wall of the keyhole was exposed to the laser beam, and the absorbed energy at the rear wall was mainly absorbed by plasma radiation and multiple reflection of the laser. The depth of the keyhole was mainly determined by the drilling behavior caused by the first absorption of laser energy at the front wall of the keyhole. The molten pool flow around the keyhole and the behavior of the keyhole were studied by some scholars [40,41,42]. The surface tension is considered to be the main driving force for the molten pool flow. The reduction of the size at the entrance of the keyhole leads to the increase of the shear stress of the steam, which accelerates the formation of spatter. The bubbles are generated by the collapse of the front and rear walls of the keyhole. Based on the pressure balance of the keyhole, Huang et al. [43] analyzed the relationship between the steam plume and the keyhole fluctuation, and believed that the change of the total pressure led to the fluctuation of the keyhole size, and the fluctuation of the plume led to the fluctuation of the hydrodynamic pressure in the keyhole.

At present, many scholars have studied the keyhole and generally agree that the formation and maintenance of the keyhole is the result of the combined effect of energy balance and pressure balance inside the keyhole. The study of energy balance in the keyhole is relatively comprehensive. Due to the keyhole being hidden in the weld pool during deep penetration laser welding, and the pressure in the keyhole being difficult to measure, most of scholars analyze and judge the pressure balance result in the keyhole on the basis of the energy balance, or conduct theoretical simulation research according to empirical formulae, so the conclusions obtained may lack the support of actual experimental data. Several experimental observation methods of keyhole research have been summarized in ref. [44]. Numerical simulation research is mainly aimed at the parameters that are difficult to detect in actual experiments, such as temperature field, pressure distribution, molten pool flow, etc., but the final results need to be confirmed by experimental results.

Limited by the existing technical means, it is difficult to directly detect the pressure inside the keyhole during deep penetration laser welding of metals, which results in difficulties in analyzing the influence of the pressure inside the keyhole and the physical properties of the material on the formation and maintenance of the keyhole. During real deep penetration laser welding of metal, a Gaussian laser can be used to irradiate the workpiece, and a keyhole surrounded by molten metal forms inside the metal. The formation and maintenance of the keyhole depends on pressure balance. When a convergent nozzle is used, a gas jet with a Gaussian velocity distribution can form. If the gas jet is directly used to impact on the liquid, under certain conditions, a keyhole can be formed in the liquid. The formation and maintenance of the keyhole are also related to pressure balance. In this paper, the authors attempted to develop an effective experimental method to study the influence of pressure on the keyhole shape by means of producing an analogy keyhole in liquid materials by blowing focused high-speed gas. So, an analogy method of keyhole formation, namely a keyhole formed by a gas jet impinging on liquid, was used to simulate the keyhole in deep penetration laser welding. Both the analogy welding and real deep penetration laser welding were carried out on liquid and a modified sandwich structure, respectively. In the analogy welding, liquid was used to simulate the molten pool, and the gas jet was used to simulate the vapor. In order to be as close as possible to the real molten metal material in welding, several materials were used in the analogy welding, including liquid alloy material at normal temperature. The behaviors of the keyhole in analogy welding and real deep penetration laser welding were intuitively observed, respectively, and the influences of relevant parameters on the behavior of the keyhole explored. If the behavior of the keyhole, such as keyhole shape, was similar in analogy welding and real deep penetration laser welding, it might offer a way to study the relationship between the pressure in the keyhole, together with the physical properties of the material and the keyhole shape, in deep penetration laser welding, instead of directly detecting the pressure in the keyhole.

## 2. Materials and Methods

### 2.1. Interaction between Gas Jet and Liquid

The flow formed by gas jetting from the orifice, nozzle or slit is called the gas jet. In this experiment, the fundamental reason for the formation of the keyhole was the interaction between the gas jet and liquid. A keyhole can form when high speed gas impinges on a liquid surface. The surface tension of the liquid is tangent to the liquid surface at the boundary and has the ability to shrink the liquid surface. The surface tension *T* can be expressed as [45]:(1)$$T=\sigma L_{b}$$
where *σ* is the surface tension coefficient, and *L_b_* is total length of boundary.

Due to the existence of surface tension, when the liquid surface is no longer flat, the additional pressure ∆*P* on the liquid surface caused by the surface tension can be expressed as [46]:(2)$$\Delta P=\sigma \left(\frac{1}{R_{1}}+\frac{1}{R_{2}}\right)$$
where *R*_1_ and *R*_2_ are the radius of curvature of the liquid surface, which is positive when the liquid surface is convex and negative when the liquid surface is concave.

The comprehensive pressure *P* on the gas–liquid interface can be expressed as [47,48]:(3)$$P=P_{a}+P_{d}+\Delta P+P_{s}$$
where *P_a_* is atmospheric pressure, *P_d_* is dynamic pressure generated by gas jet, ∆*P* is additional pressure caused by surface tension, *P_S_* is hydrostatic pressure at the position of liquid surface.

The study of a gas jet impinging on liquid is common in the metallurgical industry. The diagram of the keyhole formed by the gas jet impinging on the liquid surface is shown in Figure 1. In the figure, a rectangular coordinate system *O-xy* was established; *p*_0_, *T*_0_ and *ρ*_0_ are, respectively, the pressure, temperature and density in the nozzle, while *p*_1_, *T*_1_ and *ρ*_1_ are, respectively, the pressure, temperature and density at the nozzle outlet. The temperature and density of the jetted gas are *T_g_* and *p_g_*, respectively. The distance between the nozzle outlet and the liquid level is *H*, the keyhole depth is *h*_0_, the radius of curvature at the depth *h* on the keyhole wall is *R*. The normal of a point on the keyhole wall is *n*, the outlet diameter of the nozzle is *D*.

In many studies of the gas jet–liquid system, the keyhole depth *h*_0_, dependent on the jet action, was derived. The balance equation on the keyhole wall can be expressed as [49]:(4)$$\frac{\rho_{g}v_{g}^{2}}{2}=\rho_{L}gh_{0}+\frac{2\sigma_{L}}{R_{kh}}$$
where *ρ_g_* is the gas density at the liquid surface, *v_g_* is the velocity of the gas in the jet prior to the collision with the surface of the liquid, *ρ_L_* and *σ*_L_ are the density and the surface tension coefficient of the liquid, *h*_0_ is the keyhole depth, *R_kh_* is the radius of curvature of the liquid surface at the keyhole depth *h*_0_.

### 2.2. Pressure Balance in the Keyhole Formed in Laser Welding

In the interaction between the gas jet and the liquid, the hydrodynamic pressure generated by the Marangoni effect and eddy current are ignored. In the following study, it was assumed that the hydrodynamic pressure was ignored, and then the comprehensive pressure *P_w_* on the keyhole, when maintaining dynamic balance in a quasi-steady state, could be expressed as [50]:(5)$$P_{w}=P_{a}+P_{v}+\Delta P+P_{s}$$
where *P_v_* is the vapor pressure of the material.

### 2.3. Experimental Setup

The keyhole behavior was observed by using the experimental setup described in Figure 2 in both analogy welding and real deep penetration laser welding with the sandwich method. In Figure 2a, the experimental setup was mainly composed of a nozzle, an acrylic container, a high-speed camera and corresponding instruments. The acrylic container contained liquid materials. The high-speed camera was placed on the side of the container, and its lens direction was perpendicular to the moving direction of the nozzle. Argon was used as the gas source in the experiment. After passing through the reduction valve and the gas flow controller, the gas inside the gas source was ejected from the nozzle outlet at a certain flow rate, impacting on the surface of the liquid and forming a keyhole therein. In addition, the nozzle was fixed on an ABB Robot to realize the movement. The high-speed camera used in the experiment was a NAC MEMRECAM HX-7S.

In order to analyze the influence of different material properties on keyhole behavior, water, NaCl solution and an alloy of Ga–In–Sn were selected as the liquid materials in the experiment. The viscosity and surface tension coefficients of the materials are shown in Table 1. The variation of surface tension coefficient of NaCl solution with concentration is shown in Figure 3.

In Figure 2b, the experimental setup was mainly composed of transparent heat-resistant glass, a workpiece, a high-speed camera and corresponding instruments. The glass and workpiece were put together and clamped by a clamp. The high-speed camera was placed on the side of the workpiece, and several attenuation elements were installed on the lens to reduce the light intensity and avoid damaging the photosensitive elements inside the high-speed camera. A fiber laser was employed in the experiment, model IPG YLS-4000-CL. The workpiece was made of SUS 304.

### 2.4. Procedure

During the experiment in analogy welding, the gas flow was regulated by the gas flow controller. When the initial velocity of the gas jet changed, keyholes with different shapes formed in the liquid. The experiments included the following components:(1)The liquid surface was suddenly impacted by the gas jet, and the formation process of the keyhole recorded by the high-speed camera. The gas flow changed, and the change of keyhole depth when the nozzle was stationary was recorded.(2)The nozzle moved horizontally relative to the container at a certain speed *v* to simulate the laser welding process. The corresponding parameters of analogy welding (moving speed, gas flow rate, distance between nozzle outlet and liquid level, etc.) changed, and the behavior of the formed keyhole was recorded.(3)Different liquid materials were used, and the above experimental process was repeated for each, and the behavior of the keyhole recorded.(4)During the experiment in real deep penetration laser welding with the sandwich method, the position of the laser head was adjusted so that the center of the laser beam was located at the interface between the workpiece and the glass. The laser beam was moved at a constant speed along the interface, and the keyhole behavior in welding recorded.

## 3. Results and Discussion

### 3.1. Keyhole Formation Process

In this process, the nozzle was stationary, the gas flow rate was 82 mL/min, the diameter of nozzle outlet was 0.2 mm, and the distance between nozzle outlet and liquid level was 5 mm. The adopted high speed camera frame rate was 10,000 fps. The gas velocity *V* at the nozzle outlet depended on the gas flow rate. The gas velocity *V* could be expressed as [57]:(6)$$V=\frac{4q}{\pi D^{2}}$$
where *q* is the gas flow rate, and *d* is the diameter of nozzle outlet.

The formation of the keyhole in analogy welding is shown in Figure 4. It can be seen that under the action of the gas–liquid two-phase flow, at first, a shallow pit formed in the liquid surface, as shown in Figure 4a,b, and then a keyhole was drilled in the downward direction, as shown in Figure 4c–h, and, finally, a certain keyhole depth was reached and maintained, as shown in Figure 4i–l.

When the nozzle was stationary, the variation of keyhole depth with time or gas velocity is shown in Figure 5. Figure 5a shows the variation of keyhole depth with time in Figure 4. It can be seen that during the keyhole formation process, the change of keyhole depth had the characteristics of rapid increase to gradual stabilization, and the keyhole depth would eventually be maintained near a certain value. According to the variation of keyhole depth, the formation process of keyhole in the experiment could be divided into three stages:(1)Fast-drilling stage.(2)Slow-drilling stage.(3)Quasi-steady state stage.

In the fast-drilling stage, the keyhole depth increased rapidly, and the drilling rate could reach 320 mm/s. When the keyhole depth approached the maximum value, it entered the stage of slow-drilling, at which time the change rate of keyhole depth gradually decreased. Then it entered the quasi-steady state stage, and the keyhole depth was dynamically maintained around a certain value. There were many similarities in behavior between the keyholes observed in the analogy welding and the keyholes obtained in the real laser welding process [25,37,38].

From Figure 5b, it can be seen that the keyhole depth was positively correlated with the gas velocity. In this experiment, the dynamic pressure of the gas jet was the driving force for the formation of the keyhole, which was similar to the effect of the vapor pressure generated by the evaporation of materials in laser welding. Gas flow rate used to analogize laser power may be of significance to the study of laser welding.

At the initial stage of keyhole formation, the dynamic pressure of the gas jet was greater than the surface tension and hydrostatic pressure, so the keyhole depth changed significantly at this stage. With increase of keyhole depth, the gas dynamic pressure directly acting on the liquid surface also decreased, while the hydrostatic pressure increased. Moreover, the curvature radius of the curved surface at the bottom of the keyhole decreased, and the additional pressure caused by the surface tension increased. Therefore, the growth rate of the keyhole depth gradually decreased, and, finally, the keyhole depth oscillated near an equilibrium point.

### 3.2. Keyhole in Analogy Welding

In analogy with laser welding, the corresponding parameters (moving speed, gas flow rate, the distance between nozzle outlet and liquid level, etc.) were considered.

#### 3.2.1. Effect of Moving Speed on Keyhole

The horizontal moving speed of the nozzle was set, while other parameters remained unchanged. The obtained keyhole shapes, at different moving speeds, are shown in Figure 6. It can be seen that when the nozzle moved horizontally, the shape of the keyhole bent to different degrees with the change of the moving speed of the nozzle, and the bending direction was the opposite direction of the moving speed.

A bending keyhole is shown in Figure 7a. The angle between the keyhole drilling speed and keyhole moving speed was used to define the bending angle of the keyhole. The tangent value of angle *θ* was used to characterize the bending degree of the keyhole, and the larger the value, the more curved the keyhole was. The tangent of angle *θ* was expressed as:(7)$$\tan\theta =\frac{v_{h}}{v_{d}}$$
where *v_h_* and *v_d_* are the moving speed and drilling speed of the keyhole, respectively.

According to Formula (2), it can be seen that the shape of the keyhole was directly affected by the convexity–concavity of the keyhole wall. In the horizontal direction, the keyhole was always forced to close by the surface tension, while in the vertical direction, due to the change of the keyhole curvature, the effect of the surface tension also changed.

In Figure 7b the bending keyhole was divided into the top part, the middle part and the bottom part.

The part with a smaller radius of curvature at the keyhole inlet is marked as the keyhole top part, the part in the middle of the keyhole, where it is relatively smooth in the vertical direction, is marked as the keyhole middle part, and the concave liquid surface at the bottom of the keyhole is marked as the keyhole bottom part. The resultant force directions of the surface tension in the transverse direction and the longitudinal direction are indicated by *F_H_* and *F_L_*, respectively.

At the keyhole top part, the radius of curvature of the front and rear walls of the keyhole differed, as the surface tension kept the keyhole open in the longitudinal direction, and tended to close the keyhole in the transverse direction. A dynamic balance formed, due to the surface tension, dynamic pressure and hydrostatic pressure. The radius of curvature of the keyhole front wall in the longitudinal direction was small, and the effect of surface tension significant. This was because the gas jet directly acted on the front wall during the keyhole motion, resulting in greater gas dynamic pressure, while the larger radius of curvature of rear wall was related to the flow characteristics of the liquid.

At the keyhole middle part, the front and rear wall differed greatly. From the longitudinal perspective, the direct effect of the dynamic pressure on the front wall was relatively significant, which increased the liquid flow velocity along the wall of the keyhole. The wall had a large radius of curvature and was concave in shape. The longitudinal resultant force of surface tension had a closing effect on the keyhole, but the effect was not obvious. However, a part of a convex shape appeared in the rear wall, as the effect of the surface tension in the longitudinal direction here was the same as that of the gas dynamic pressure to maintain the keyhole opening.

At the keyhole bottom part, the shape was concave, the curvature radius of the keyhole was small, and the effect of surface tension was significant. The gas pressure was the main driving force to maintain the keyhole opening, while the hydrostatic pressure and surface tension tended to close the keyhole.

As shown in Figure 6, it can be seen that, at different moving speeds, the keyhole appeared to have different bends, and so was the keyhole depth. The effect of the change of moving speed on the keyhole depth is shown in Figure 8a. It can be seen that the keyhole depth decreased with increase of the nozzle moving speed, and the relationship between the two was almost linear. The relationship between the bending degree of the keyhole and the moving speed of the nozzle is shown in Figure 8b. The result shows that the bending degree of the keyhole increased with the moving speed of the nozzle, and the relationship between them was also approximately linear. It was obvious that the nozzle moving speed had a linear relationship with the keyhole depth and keyhole bending degree, which indicated that there was a certain correlation between the keyhole depth, keyhole bending degree and the nozzle moving speed. The effect of nozzle moving speed variation in the analogy welding was similar to the effect of welding speed variation in real deep penetration laser welding.

#### 3.2.2. Effect of Distance between Nozzle Outlet and Liquid Level on Keyhole

The nozzle was stationary, the gas flow rate was 81.5 mL/min, the distances between nozzle outlet and liquid level were 3 mm, 4 mm, 5 mm, 6 mm and 7 mm, respectively. The variation of keyhole depth with time is shown in Figure 9. It can be seen that when the distance reduced, the keyhole drilling rate in the fast-drilling stage also increased, but when the quasi-steady state stage was reached, the fluctuation of the keyhole would also become more violent. The change of the distance between nozzle outlet and liquid level was actually the change of the gas velocity on the liquid surface and the change in the gas–liquid interaction area. The effect of distance variation in the analogy welding was similar to the effect of defocusing variation in deep penetration laser welding.

#### 3.2.3. Effect of Material Properties on Keyhole

As shown in Figure 3, the surface tension coefficient of NaCl solution was linearly correlated with solute concentration. The surface tension coefficient is an important parameter in Formula (4). A series of NaCl solutions with different surface tension coefficients were prepared, which were impacted by certain gas flows, respectively. The obtained keyhole depth of each solution at the quasi-steady state stage with different surface tension coefficients is shown in Figure 10. Obviously, the keyhole depth was negatively correlated with the surface tension coefficient.

According to Formula (2), the effect of surface tension was related to the radius of curvature, so the smaller the radius of curvature, the greater the influence of surface tension. When NaCl solutions with different mass fractions were used in the experiment, the density changed little, and it was considered to be a constant. The surface tension had the ability to close the keyhole, by calculation, and it could be seen that the additional pressure caused by the surface tension was much greater than the hydrostatic pressure at the bottom of the keyhole, so it could be considered that the surface tension of the material was the main factor determining the maximum depth of the keyhole.

Considering that the keyhole in laser welding is surrounded by molten metal, liquid metal material was selected as the experimental material for the analogy welding. For experimental safety, the alloy of Ga–In–Sn was selected as unselected mercury. Figure 11 shows the keyhole shape in the Ga–In–Sn alloy and the keyhole is outlined with red line. The keyhole in Figure 11a was obtained when the gas flow was 300 mL/min and the distance between nozzle outlet and liquid level was 3 mm. It can be seen that when the selected material physical properties were relatively close to those of the real molten pool material, the keyhole shape obtained by the analogy welding was similar to an inverted cone with a wide upper part and a narrow lower part.

When the nozzle was moved at a horizontal speed of 30 mm/s, the keyhole obtained is shown in Figure 11b. The front wall of the keyhole bent in the opposite direction of the moving speed, and the bending degree increased with increase in the moving speed.

When the distance between nozzle outlet and liquid level changed, the drawn curve of the maximum depth of the keyhole is shown in Figure 12. The effect of distance variation on keyhole depth in liquid metal was similar to the effect of defocusing variation in laser processing.

The properties of liquid metal Ga–In–Sn alloy were similar to those of molten metal in welding. It may be of reference significance to explore the keyhole formed by gas blowing in analogy welding to study the keyhole in laser deep penetration welding.

### 3.3. Keyhole in Real Deep Penetration Laser Welding with Sandwich Method

The keyhole behavior in real deep penetration laser welding with the sandwich method was captured by the high-speed camera. The adopted camera frame rate was 10,000 fps, and several attenuation elements were used. The set laser power was 2500 W and the defocusing was 0 mm. Different welding speeds were set, and the obtained keyholes are shown in Figure 13. It can be seen that all keyholes at different welding speeds were bent in the opposite direction of the welding speed. A straight line passes through the top point of the keyhole front wall and is tangent to the keyhole front wall. The angle *θ* between that straight line and the vertical line is defined as the bending angle of the keyhole. The relationship between welding speed and keyhole depth and keyhole bending angle is shown in Figure 14. Comparing Figure 8 with Figure 14, it can be seen that the variation rules of keyhole depth and keyhole bending were basically the same in the analogy welding and in real welding. During a welding process, the variation of keyhole depth is shown in Figure 15. Comparing Figure 15 with Figure 5a, it can be seen that the variation trend of keyhole depth was similar. The stages of keyhole depth change in Figure 15 could still be divided into fast-drilling stage, slow-drilling stage and quasi-steady state stage.

Figure 16a shows the keyhole obtained in real deep penetration laser welding when the defocusing was 5 mm, and Figure 16b shows the keyhole in analogy welding when the distance between nozzle outlet and liquid level was 5 mm. It can be seen that the two keyholes were very similar, both of them being slightly slender in shape. The front and rear walls in the keyhole middle part were smooth, while the bottoms of the two keyholes were curved and semicircular. In addition, the dynamic behavior of the two keyholes was similar. For example, the keyholes bent in the opposite direction of the welding speed, and the faster the speed was, the greater the bending degree was, and the smaller the keyhole depth was.

The main driving force of the keyhole generated by deep penetration laser welding is the vapor pressure generated by material evaporation of laser irradiated materials, while the force generating the keyhole in the analogy welding is mainly the dynamic pressure generated by the gas jet. Comparing Formula (3) with Formula (5), it can be seen that the composition of comprehensive pressure in the two kinds of keyholes is similar in form despite that the physical meanings being different. Mechanical properties influence the characteristics of keyholes. There are many similarities in the shape and dynamic behavior of keyholes obtained by analogy welding and real deep penetration laser welding, and the influence of process parameters on them is also similar. The study of keyhole formation and the maintenance mechanism obtained by analogy welding is helpful to understand the relevant mechanism in the real deep penetration laser welding process.

## 4. Conclusions

In this paper, the keyholes in analogy welding and deep penetration laser welding were directly observed, and the effects of experimental parameters and material properties on the behavior of the keyholes were studied. The keyhole formed by gas blowing was compared with the keyhole generated in deep penetration laser welding. The following conclusions could be made:(1)In the analogy welding and real deep penetration laser welding with the sandwich method, the pressure balance on the keyhole is similar in form. The gas dynamic pressure acting on the liquid in analogy welding and the vapor pressure caused by the evaporation of the laser irradiated material in deep penetration laser welding both promote the opening of the keyhole, while the hydrostatic pressure and surface tension promote the closing of the keyhole.(2)When the process parameters in the analogy welding and real laser welding are used to study the influence on the keyhole, the behavior of the keyhole in the analogy welding is similar to that obtained in the real deep penetration laser welding.(3)Studying the keyhole formation and maintenance principle in analogy welding may be helpful to deeply understand the keyhole formation and maintenance mechanism in real deep penetration laser welding.

## Figures and Tables

**Figure 1 materials-15-09001-f001:**
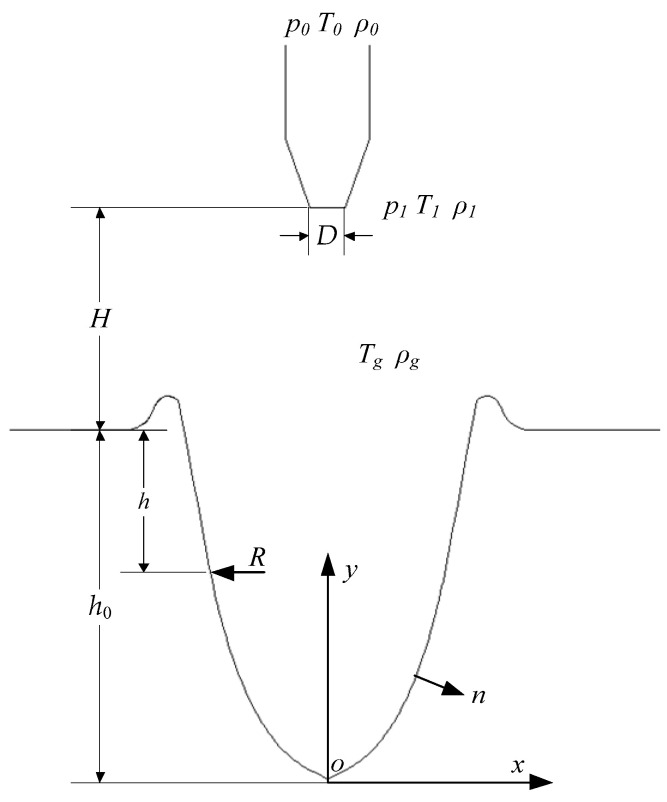
Diagram of the keyhole formed by a gas jet impinging on the liquid surface.

**Figure 2 materials-15-09001-f002:**
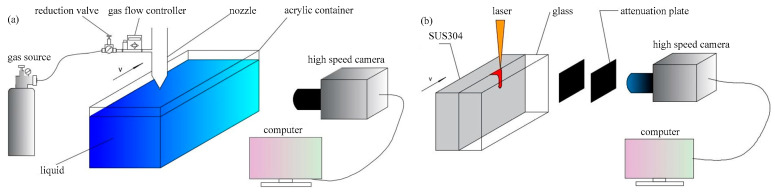
Diagram of the keyhole behavior observation experimental setup: (**a**) Keyhole observation in analogy welding; (**b**) Keyhole observation in real deep penetration laser welding with sandwich method.

**Figure 3 materials-15-09001-f003:**
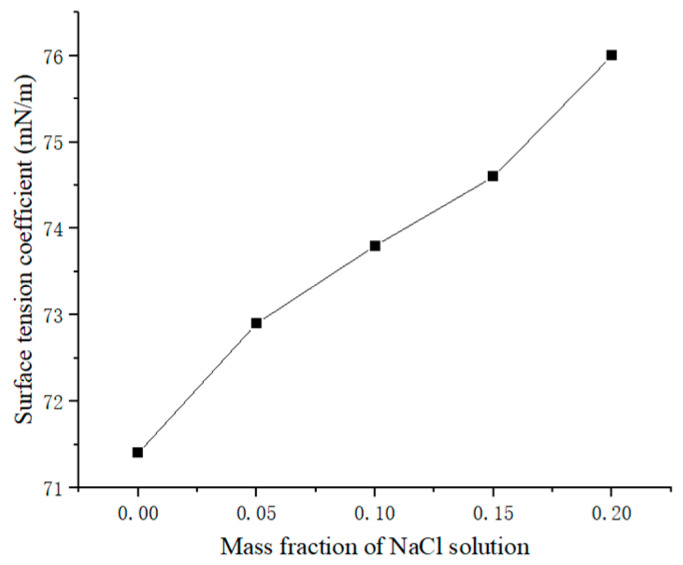
Variation of surface tension coefficient of NaCl solution with mass fraction.

**Figure 4 materials-15-09001-f004:**
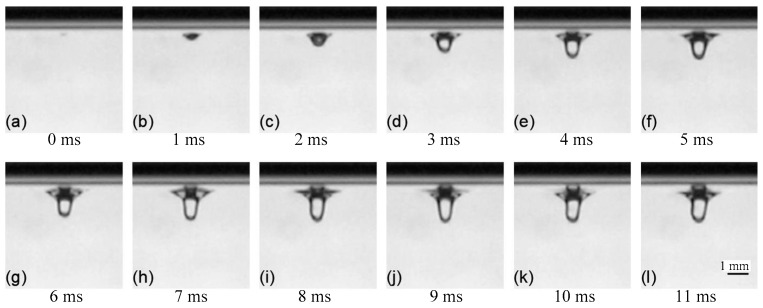
Formation of keyhole in analogy method: (**a**) No keyhole formed at 0 ms; (**b**) Keyhole at 1 ms; (**c**) Keyhole at 2 ms; (**d**) Keyhole at 3 ms; (**e**) Keyhole at 4 ms; (**f**) Keyhole at 5 ms; (**g**) Keyhole at 6 ms; (**h**) Keyhole at 7 ms; (**i**) Keyhole at 8 ms; (**j**) Keyhole at 9 ms; (**k**) Keyhole at 10 ms; (**l**) Keyhole at 11 ms.

**Figure 5 materials-15-09001-f005:**
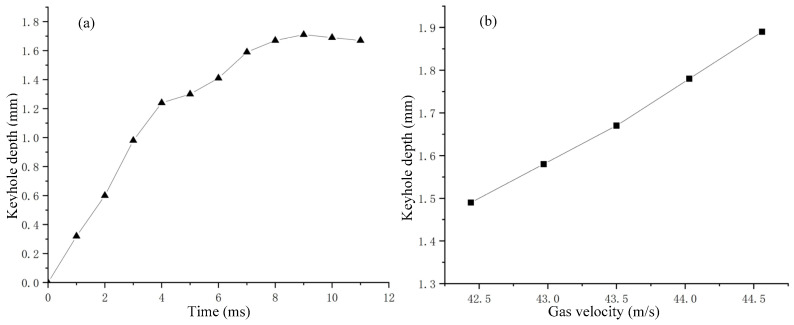
The variation of keyhole depth: (**a**) Keyhole depth with time; (**b**) Keyhole depth with gas velocity.

**Figure 6 materials-15-09001-f006:**
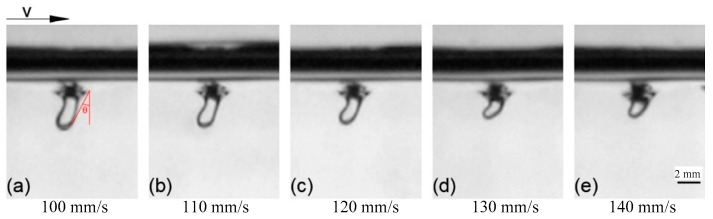
Keyhole during horizontal movement of nozzle: (**a**) Keyhole with nozzle moving speed of 100 m/s; (**b**) Keyhole with nozzle moving speed of 110 m/s; (**c**) Keyhole with nozzle moving speed of 120 m/s; (**d**) Keyhole with nozzle moving speed of 130 m/s; (**e**) Keyhole with nozzle moving speed of 140 m/s.

**Figure 7 materials-15-09001-f007:**
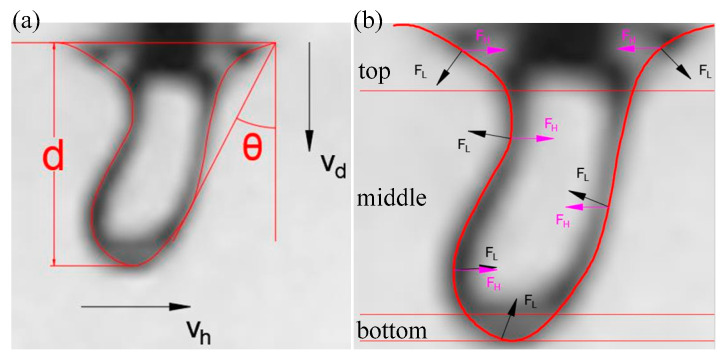
Bending keyhole: (**a**) Angle of bending keyhole; (**b**) Segmentation of bending keyhole.

**Figure 8 materials-15-09001-f008:**
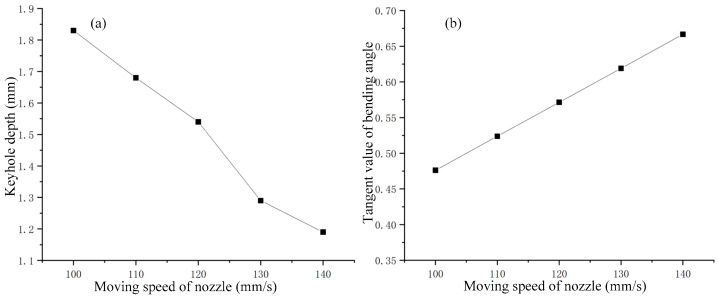
Effect of nozzle moving speed on keyhole: (**a**) Effect on keyhole depth; (**b**) Effect on keyhole bending.

**Figure 9 materials-15-09001-f009:**
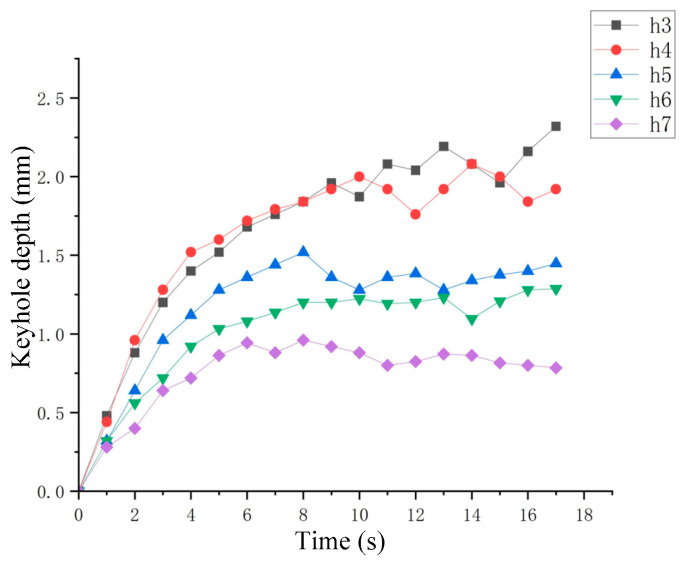
Effect of distance between nozzle outlet and liquid level on keyhole depth.

**Figure 10 materials-15-09001-f010:**
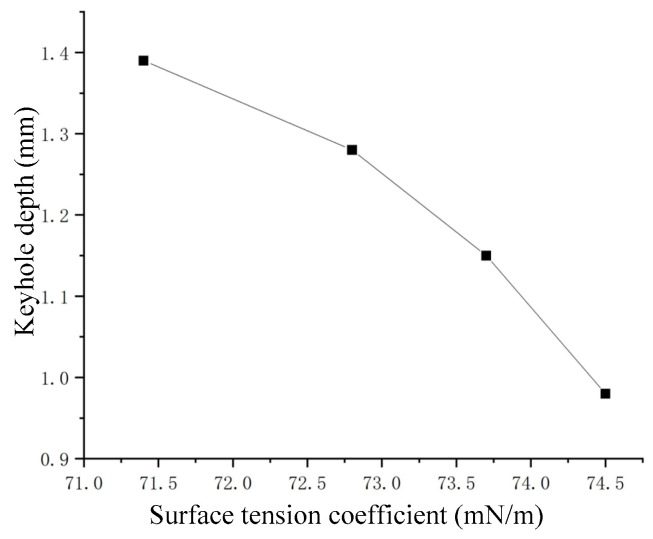
Keyhole depth corresponding to different surface tension coefficients.

**Figure 11 materials-15-09001-f011:**
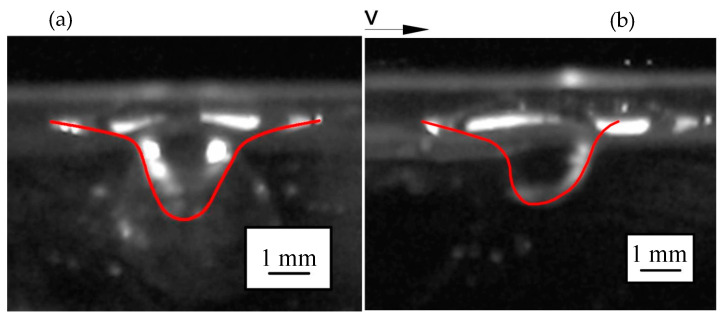
Keyhole in Ga–In–Sn alloy: (**a**) Stationary keyhole; (**b**) Dynamic keyhole.

**Figure 12 materials-15-09001-f012:**
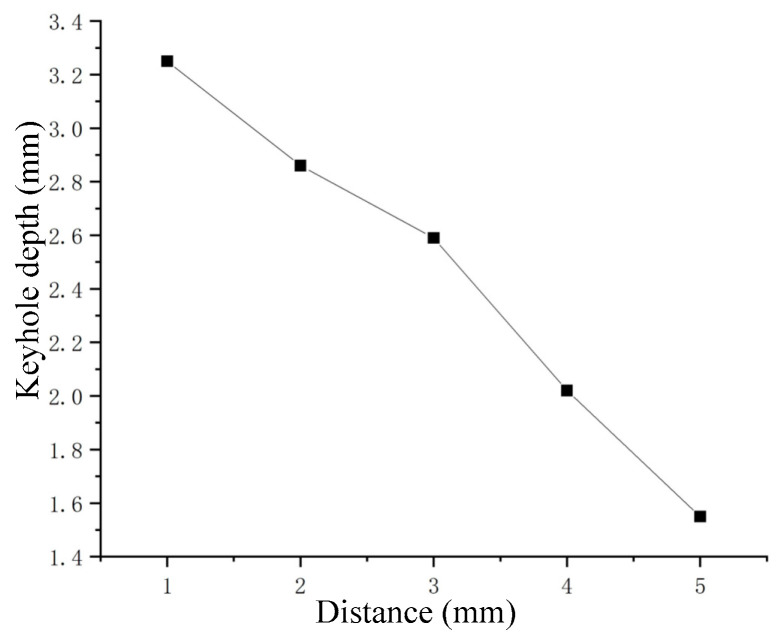
Keyhole depth in Ga–In–An alloy corresponding to different distance of nozzle.

**Figure 13 materials-15-09001-f013:**
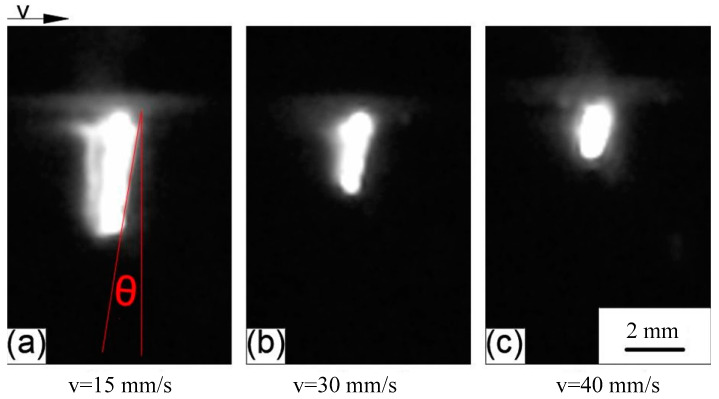
Keyhole shape at different speeds observed by the sandwich method: (**a**) Keyhole shape at welding speed of 15 mm/s; (**b**) Keyhole shape at welding speed of 30 mm/s; (**c**) Keyhole shape at welding speed of 40 mm/s.

**Figure 14 materials-15-09001-f014:**
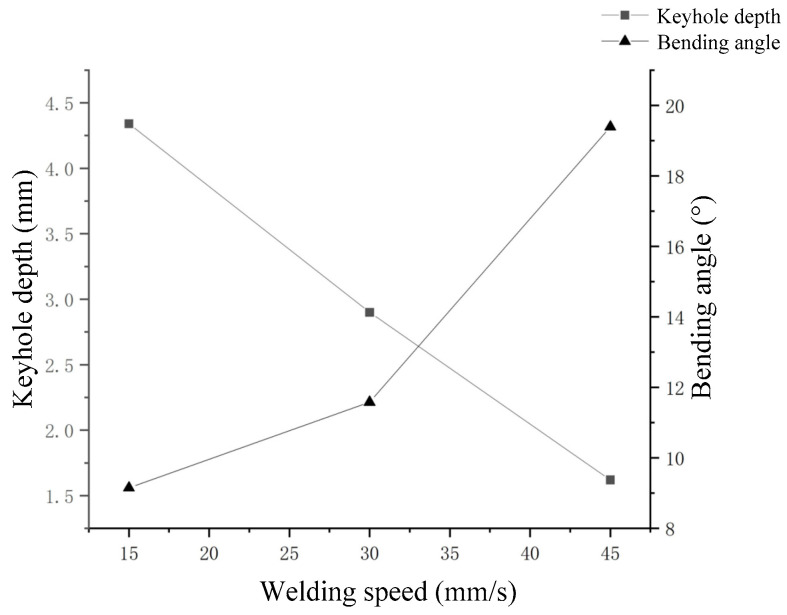
Effect of welding speed on keyhole depth and bending angle.

**Figure 15 materials-15-09001-f015:**
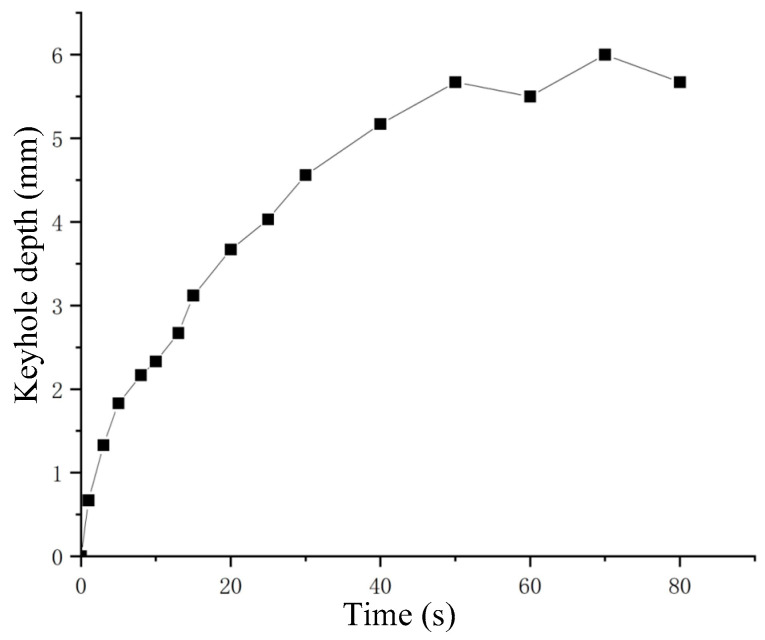
The variation of keyhole depth with time.

**Figure 16 materials-15-09001-f016:**
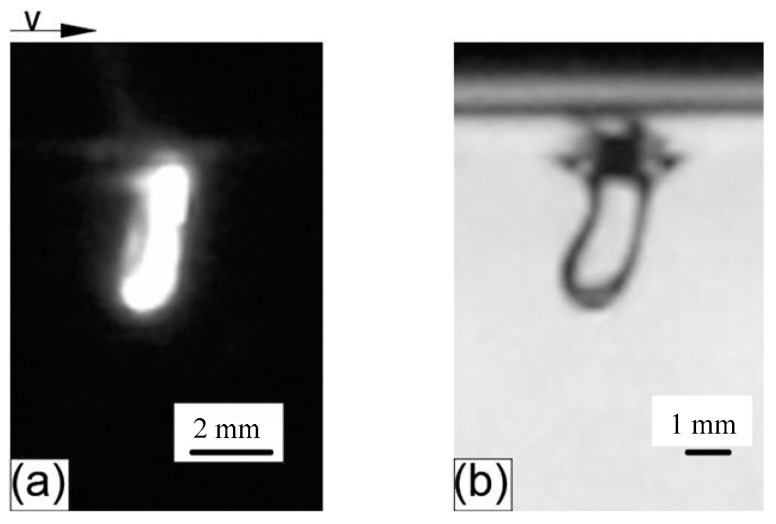
Obtained keyhole: (**a**) Keyhole in real deep penetration laser welding; (**b**) Keyhole in analogy welding.

**Table 1 materials-15-09001-t001:** Viscosity and surface tension coefficient of materials [51,52,53,54,55,56].

Material	Viscosity (cp)	Surface Tension Coefficient (mN/m)
Water	1	71.4
NaCl solution	1~2	71.4~78.7
Ga-In-Sn alloy	2.4	718

## Data Availability

The data that support the findings of this study are available upon reasonable request from the authors.

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
