# Peer review of "Comparative Study on the Behavior of Keyhole in Analogy Welding and Real Deep Penetration Laser Welding"

_materials, 2022, doi:10.3390/ma15249001_

Round 1
Reviewer 1 Report
Paper is devoted to the study of the mechanism laser welding. The authors of paper attempts to explain the process of material movement and keyhole formation during laser melting using model experiments. There are no remarks from the point of view of the methodology of the experiments performed.
But the main goal of the work is unclear. The conclusions of the paper state that formulas 3 and 5 are similar, this statement is devoid of any physical meaning. The recoil pulse during laser evaporation strongly depends on the power density of the laser radiation. This fact is confirmed by nonlinear distortions of the keyhole in experiments with sandwich structure. The paper does not establish a clear similarity between the behavior of the liquid under the gas jet and the laser melting of the material and the formation of the keyhole.
The authors need to clearly formulate the goal of the work and tasks, without this the work has no relevance. To date, there are works on mathematical modeling of the laser welding process that converge well with experiments. Therefore, the purpose and relevance of the presented work is completely unclear.
Author Response
Response: Thank you for your comment and suggestion. Limited by the existing technical means, it is difficult to detect directly the pressure inside the keyhole during deep penetration laser welding of metals, which brings difficulties for analyzing the influence of the pressure inside the keyhole and the physical properties of the material on the formation and maintenance of the keyhole. In this manuscript, we attempt to seek an effective experimental method to study the influence of the pressure in the keyhole on the keyhole shape by means of producing an analogy keyhole in liquid materials by blowing focused high speed gas, which can build the relationship between the pressure in the keyhole together with the physical properties of the material and the keyhole shape in deep penetration laser welding based on similar laws of keyhole shapes, instead of directly detecting the pressure in the keyhole.
During real deep penetration laser welding of metal, a Gaussian laser can be used to irradiate the workpiece, and a keyhole surrounded by molten metal will form inside the metal. The formation and maintenance of keyhole are related to pressure balance.
When a convergent nozzle is used, a gas jet with a Gaussian velocity distribution can be formed. Inspired by this, we use the formed gas jet and liquid to form a keyhole. In the process of gas impacting liquid to form a keyhole, the formation and maintenance of keyhole are also related to pressure balance.
In this manuscript, we compare the pressure balance of the keyhole with the formulas in the two cases of analogy welding and real deep laser welding. We adopt the technological experiment and compare the two kinds of keyhole shapes in the experiments in the following ways:
- In section 3.1, keyhole formation process in liquid under gas jet was recorded in Figure 5(a), and the change process of keyhole depth was compared with that in Figure 15 in real deep laser welding in section 3.3.
- In Figure 5(b), the gas flow rate with Gaussian distribution was used to analogize the laser power with Gaussian distribution, and the keyhole depth variation was recorded. It may be of significance to the effect of power in laser welding.
- In section 3.2.1, the effect of moving speed on keyhole in liquid was recorded in Figure 8, and the variation of keyhole depth and bending angle were compared with those in Figure 14 in real deep laser welding in section 3.3.
- In section 3.2.2, the distance between nozzle outlet and liquid level was used to analogize the defocusing amount in real laser welding. It may be of significance to the effect of defocusing amount in laser welding.
- In section 3.2.3, the influence of material properties on keyhole depth was studied. NaCl solution and Ga-In-Sn liquid alloy were used under gas jet.
- In section 3.3, keyhole shapes in real deep penetration laser welding and gas impinging on liquid were compared, the results reveal that the keyhole shapes in the the above cases were similar.
- Formulas 3 and 5 are similar just in form, but different in physical meaning. Formulas 3 represents the pressure balance in the analogy keyhole, in which Pd is dynamic pressure generated by gas jet. However, formulas 5 represents the pressure balance in the real keyhole in deep penetration laser welding, in which Pv is the vapor pressure of the material and generated by laser beam.
As mentioned above, we compared the keyholes in analogy welding and real deep laser welding. We try to study the keyhole in the liquid, and perfect the relevant theory of invisible actual keyhole in deep penetration laser welding with similar laws.

Reviewer 2 Report
This article reviews the "Comparative study on the behavior of keyhole in analogy welding and real deep penetration laser welding". The article has an interesting innovation. It is especially useful for those who do laser welding. Some things need to be investigated further. At the beginning of the introduction, it is necessary to introduce the laser welding modes and then introduce the keyhole completely. The purpose of keyhole welding, etc. It is also necessary to pay more attention to the role of the material and, of course, the welding parameters of Leerer. For example, the optimal parameter that creates a keyhole in aluminum welding should be introduced. The advantages and limitations of keyhole welding are further discussed. Finally, a clear goal of this research should be provided. The authors have fully described the keyhole formation process from the beginning to the end of the formation. Well, what's the point? What does it help the reader to improve his work process? Do the authors suggest better modification of the keyhole in terms of geometry, etc.?
Author Response
Response: Thank you for your comment and suggestion. In the introduction, we have supplemented the introduction of welding mode and keyhole. For the purpose of keyhole welding, its application and advantages are supplemented in the paper. We introduce the deep penetration laser welding of aluminum plates with certain parameters and the limitations of deep penetration laser welding.
Limited by the existing technical means, it is difficult to experimentally analyze the influence of the pressure inside the keyhole and the physical properties of the material on the formation and maintenance of the keyhole during deep penetration laser welding metal. Therefore, we hope to find an effective method for experimental study of keyhole shape, especially the pressure in the keyhole. Inspired by the keyhole generated by metal vapor in liquid molten metal when Gaussian laser acts on metal material, we try to use gas jet to act on liquid to generate keyhole, and compared the characteristics of keyholes in the two cases of analogy welding and real deep laser welding, laying a foundation for the study of keyhole in real laser welding. So, we fully described the keyhole formation process from the beginning to the end, and try to build the relationship build the relationship between the pressure in the keyhole together with the physical properties of the material and the keyhole shape in deep penetration laser welding based on similar laws of keyhole shape.

Reviewer 3 Report
Interesting work showing the simulation of a molten metal pool during laser welding. These are basic research that requires further development and confirmation in further simulation studies using specialized software and subsequent laser remelting tests. In my opinion, the work can be published as a work showing the possibility of using this type of modeling for the preliminary assessment of the molten metal pool during laser welding
Author Response
Response: Thank you for your comment and suggestion. The purpose of our manuscript is to make a useful attempt to seek an effective experimental method to study the influence of the pressure in the keyhole on the keyhole shape by means of producing an analogy keyhole in liquid materials by blowing focused high speed gas, and then to build the relationship between the pressure in the keyhole together with the physical properties of the material and the keyhole shape in deep penetration laser welding based on similar laws of keyhole shapes, which can overcome the difficulties of directly detecting the pressure in the keyhole in real deep penetration laser welding. The analogy method proposed in this manuscript together with the corresponding results will lay the foundation for further development and confirmation in further simulation studies using specialized software and subsequent laser remelting tests.

Round 2
Reviewer 2 Report
The manuscript is well-polished and recommended for publication.
Author Response
Thank you for your comment and suggestion.